# Intensive Livestock Production Causing Antibiotic Pollution in the Yinma River of Northeast China

**Hanyu Ju [1,2]** , **Sijia Li [3]** , **Y. Jun Xu [4]** , **Guangxin Zhang [1,\*]** and **Jiquan Zhang [3,\*]**

[1]   Key Laboratory of Wetland Ecology and Environment, Northeast Institute of Geography and Agroecology, Chinese Academy of Sciences, No. 4888, Shengbei Street, Changchun 130102, China; juhanyu@iga.ac.cn

[2]   University of the Chinese Academy of Sciences, Beijing 100049, China

[3]   Institute of Natural Disaster Research, School of Environment, Northeast Normal University, Changchun 130024, China; lisj983@nenu.edu.cn

[4]   School of Renewable Natural Resources, Louisiana State University Agricultural Center, Baton Rouge, LA 70803, USA; yjxu@lsu.edu

**\***   Correspondence: zhgx@neigae.ac.cn (G.Z.); Zhangjq022@nenu.edu.cn (J.Z.); Tel.: +86-139-4481-3495 (G.Z.); +86-135-9608-6467 (J.Z.)

**Abstract:** Antibiotics are increasingly used in livestock production in rural China, raising concerns over pollution and health risk in countryside waterways. The Yinma River Basin in China's far northeast is an agriculture-dominated area mixed with a densely populated province capitol city, providing a suitable area for investigating the influence of a typical land use mix in Northeast China on riverine antibiotic levels and transport. In this study, we sampled water along the Yinma River from upstream to downstream in a wet and a dry season and analyzed the samples for two popularly used antibiotics, ciprofloxacin (CIP) and norfloxacin (NOR). The goal of the study was to determine the spatiotemporal distribution of the antibiotics in Yinma's two tributaries, Yitong and Yinma, which drain intensive livestock production land, and to elucidate which environmental and social factors influence the distribution of antibiotics in the cold and low mountainous areas. Water sample collection and instream measurements on dissolved oxygen and other ambient conditions were conducted at 17 locations along the Yinma and Yitong tributaries in August 2015 (wet season) and November 2015 (dry season). In addition to determining CIP and NOR levels, water samples were also analyzed for dissolved organic carbon (DOC), ammonia ($NH_3$), and free chlorine. We found a significantly higher level of NOR when compared to CIP, indicating greater use of the first in livestock production. The level of both antibiotics was higher in the wet season (NOR: 61.063 ± 13.856 ng $L^{-1}$; CIP: 3.453 ± 0.979 ng $L^{-1}$) than in the dry season (57.435 ± 14.841 ng $L^{-1}$; 3.091 ± 0.824 ng $L^{-1}$), suggesting higher runoff of the antibiotics from the drainage area during the raining season. The level of antibiotics was higher in rural areas, especially forested and wetland areas where livestock typically graze, as well as in the lower river basin. However, the health risk of antibiotics is determined by the physical condition and lifestyle of the residents in the river basin, hence showing a higher vulnerability of the urban area than the rural area.

**Keywords:** water quality; antibiotics; intensive livestock production; environment and social factors; health risk; Yinma River

## 1. Introduction

Large amounts of antibiotics are now being used in agriculture as growth promoters [1,2]. Especially, antibiotics are widely used in livestock production and aquaculture in the form of animal feed additives to prevent the occurrence of animal diseases [3,4]. Previous studies have shown that most of the chemical compounds are not completely metabolized and residues of the antibiotics (80–90%)



used in humans are excreted with urine and feces [5,6], reaching urban sewage treatment plants and rivers by soil leaching, where they may escape degradation and can contaminate waste, surface, and groundwater [7]. Some studies reported that antibiotic contamination can reach a nanograms per liter level [8]. With long-term exposure to this environment, the load of antibiotics could have an impact on organisms, e.g., the emergence of drug-resistant strains, inhibiting the growth of plants and animals, and causing harm to human health [9,10].

Several studies [7,11,12] indicated that the main source of antibiotic pollution in the river basins of northern China came from livestock production, aquaculture, sewage discharge, and wastewater treatment plant drainage. In livestock fields, antibiotics can enter the natural environment through metabolites, surface runoff, soil leaching, and illegal dumping [13,14]. Discharge from aquaculture can directly transport antibiotics into natural waterways [2,11,15]. Studies have shown that domestic sewage can also contain a large number of pharmaceutical residues, which are discharged into waterways in a river basin [6]. As domestic sewage and medical wastewater all contain large amounts of antibiotics, and sewage treatment plants cannot degrade antibiotics, sewage treatment plants, which have accumulated a large number of antibiotics, have become a major source of antibiotic pollution [16]. Although the typical sources of antibiotics in the river basin have been identified, the distribution of antibiotics under the influence of a dominant factor has not been studied in depth.

Studies have found that the distribution of dissolved organic carbon (DOC) is correlated with various environmental and population economic factors [3,17]. Antibiotics as a pseudo-persistent organic pollutant exist in the natural environment for a long time and are an important part of DOC in waterways [18]. Therefore, its distribution in river water will also be affected by pollution sources and environmental and social factors, such as hydrological, climatic, topography, landform, land use, population, and economic development. However, direct exploration of the correlation between the distribution of antibiotics and various environmental and social factors is still to be studied.

This study aimed to investigate the occurrence, distribution, and health risks of two popularly used antibiotics in a headwater area in Northeast China, the Yinma River Basin. We hypothesized that intensive livestock production in headwater areas can lead to basin-wide antibiotic pollution. Specifically, the study aimed to: (1) Assess the spatiotemporal occurrence and distribution of antibiotics in tributary waters; (2) analyze the relation of the antibiotics with major environmental and social factors in the headwater basin; and (3) quantify the health risk of antibiotics based on the natural and anthropogenic conditions in the river basin.

## 2. Materials and Methods

### 2.1. Study Area

This study was conducted in the Yinma River Basin in Northeast China, which is comprised of two tributary basins: The Yitong tributary and the Yinma tributary basins (Figure 1). The entire river basin is termed as the Yinma River Basin in Chinese, probably because of the much larger drainage area of the Yinma tributary basin. Both the Yitong and Yinma tributaries flow northward almost parallel for approximately 340 and 360 km, respectively, and then join and meet the Second Songhua River 20 km after their confluence. The Yitong tributary basin is densely populated and industrialized, while the Yinma tributary basin is more rural and agriculture intensive. The entire Yinma River Basin is approximately 17,400 km$^2$, with an elevation range between 188 and 1038 m. The area is characterized as a cold temperate monsoon region, with a long-term average annual temperature of 5.7 °C, fluctuating from −15.7 °C in January to 22.9 °C in July. Precipitation in the region is mainly concentrated during June and September (wet season), with the maximum rainfall of 866.6 mm occurring in August and the minimum annual precipitation of 329.7 mm occurring in the dry season, which is before the freeze-up period (November) [7]. The annual variation coefficient of runoff in the Yinma River Basin is 1.30, which is mainly concentrated in July to August (wet season) as 47.3 mm, while the lowest runoff is 5.4 mm in November (except the freezing season).

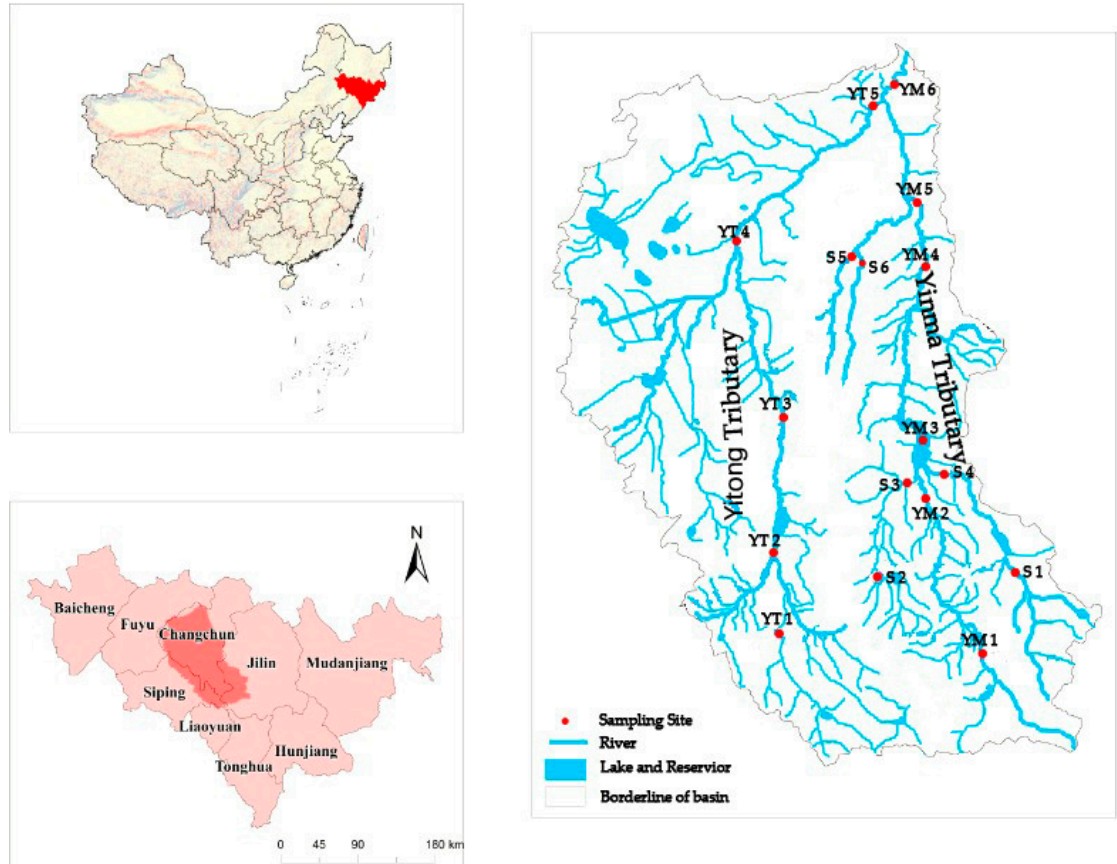

**Figure 1.** Geographical locations of the Yinma River Basin (124°58′–126°24′ E, 43°02′–44°53′ N) in Northeast China and 17 sampling sites along the Yiton Tributary in the east and the Yinma Tributary in the west (modified from the map on p. 65 by Sun et al., 2016) [19].

The Yinma River Basin has a population of nearly 9 million, most of which is centered in Changchun, the capital city of Jilin province. Agriculture is the dominant land use, followed by forestry and urban (Figure 2). The main soil types in the river basin are cambisols and phaeozems (FAO World Reference Base for Soil Resources). Pollution sources in the river basin include agricultural, industrial, and residential. Ciprofloxacin (CIP) and norfloxacin (NOR) are effective and common treatments for respiratory and digestive diseases that are common and fatal in animal husbandry. Previous studies reported higher levels of ciprofloxacin (CIP) and norfloxacin (NOR) in the river, which has been attributed to the agricultural and residential area [9].

The environmental and social factor data of the Yinma River Basin are shown in Figure 2. The slope data was obtained in digital elevation mode (DEM) data, and was extracted by 3D Analyst tool in Arc GIS 10.0 analysis software (the DEM data is used in this article are from the Shuttle Radar Topography Mission (SRTM) data set, which was jointly issued by NASA and NIMA. Each latitude and longitude grid of the SRTM data provides a file, divided into 1 arc-second (srtm-1) and 3 arc-seconds (srtm-3). The data from the computer network information center, Chinese academy of sciences, China international scientific data mirror website (http://datanirror.csdb.cn)), and land-use type in the Yinma River Basin comes from the resource and environmental science data center of Chinese academy of sciences (http://www.resdc.cn); GDP distribution and population density data for the Yinma River Basin were sourced from the resources and environmental sciences data center of the Chinese academy of sciences (http://www.resdc.cn/). For the population density data, a 1 km × 1 km grid data were generated by the method of spatial interpolation; GDP distribution data generated a 1 km × 1 km raster data by space interpolation.

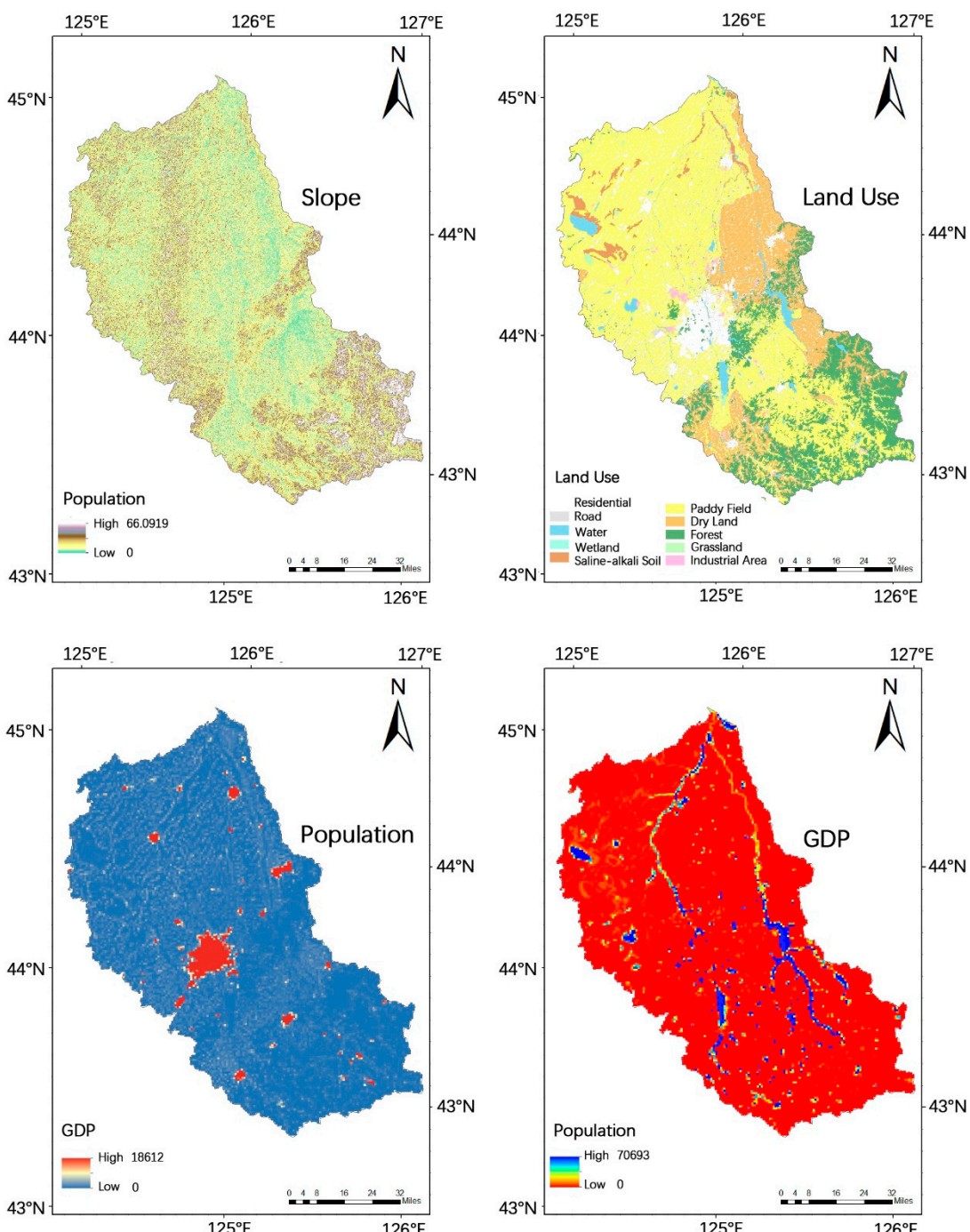

**Figure 2.** Slope in degrees (**a**), land use type (**b**), population density (person) (**c**), and gross domestic product (GDP) in Chinese currency Renminbi (RMB) distribution as an indicator for local economic development (**d**) in the Yinma River Basin, far Northeast China.

## 2.2. Sampling Design and Procedure

In this study, we selected 17 sampling sites along the Yitong and Yinma tributaries (Figure 1). Selection of the sites was to assess changes in antibiotic concentrations from the upper to the low drainage basin. A total of 5 sampling sites were located along the Yitong River mainstem, 7 sampling sites were located in the Yinma River mainstem, and the rest of the 5 sampling sites were within the two smaller tributaries of the Yinma River (Figure 1). These sampling sites spread across urban (YT3), suburb (YT1, YM5, YT4), and rural (other sites) areas. In Figure 1 (Supplementary Materials) [18],

YT3 was located at the most downstream sewage outlet in Changchun city, so this sampling site was selected to reflect the pollution situation in urban areas.

Field trips to the Yinma River were conducted during August 2015 (wet season) and November 2015 (dry season). A total of 102 water samples were collected during the two seasons. Coordinates for water sample sites were recorded using a global positional system (G350, UniSreong, China) for repeated sampling. During each trip, in-situ measurements on river water temperate, pH, and dissolved oxygen (DO) were taken. Grab waters samples were collected from the depth of 0.5 to 1 m below the water surface, by mixing three subsamples, each at least 200 m apart (to eliminate accidental interference during sampling) [20]. Samples, each approximately 20 L, were held in Perspex water sampler bottles that were cleaned with Milli-Q water and rinsed again by river water in situ before collection. The samples were stored at 4 °C in acid-cleaned pre-combusted amber bottles in the dark for laboratory analysis at the Environment Institute, Northeast Normal University in Changchun, Jilin. Physical and chemical parameters were determined within 6 h of sample collection.

## 2.3. Antibiotic Extraction and Analysis

The water samples were filtered through a 0.45-μm glass fiber filter (baked at 450 °C for 4 h) membranes, and transported to a lab for analysis within 24 h [7]. A 10-L sample of filtered water was added to 50 mL of 0.1 M Ethylenediaminetetraacetic acid (EDTA)- McIlvaine buffer and 2.4 mL of formic acid [20], followed by solid phase extraction.

Solid phase extraction was performed using Oasis HLB extraction cartridges ($0.2\,g \cdot 6\,mL^{-1}$) (Waters Corp., Milford, MA, USA). The sample and supernatant were loaded into an Oasis HLB cartridge that had been preconditioned with 5 mL of a methanol and ethyl acetate mixture (1:1 v/v), followed by 5 mL of distilled water, at a rate of $2\,drops \cdot s^{-1}$. The cartridge was kept wet or damp during the loading of the water sample, which was then washed with 6 mL of methanol (5%) at a rate of $1\,drop \cdot s^{-1}$, and the eluate was discarded. Finally, the target fraction was eluted with 5 mL of methanol at a rate of $1\,drop \cdot s^{-1}$. The eluate was reduced to 1 mL under a gentle stream of high-purity nitrogen gas, depending on the loading volume of the sample, and then transferred into a 2 mL polypropylene vial for high-performance liquid chromatography (HPLC) analysis [2].

The two popular antibiotics, norfloxacin (NOR) and ciprofloxacin (CIP), were quantified with HPLC (Waters, 2489, America). NOR concentration in the water samples was identified with a fluorescence detector, while the column was maintained at 30 °C. The mobile phase consisted of eluent A (acetonitrile) and eluent B (0.1% phosphoric acid in ultrapure water, PH = 3), and the rate ratio of these eluents was maintained at 13:87. The flow rate was maintained at $1\,mL \cdot min^{-1}$, and the injection volume was 1.0 μL. The excitation and emission wavelengths were 280 and 450 nm, respectively. The CIP concentration in the water samples was identified with a UV detector, and the column was again maintained at 30 °C. The mobile phase consisted of eluent A (acetonitrile) and eluent B ($2\,mmol \cdot L^{-1}$ of potassium dihydrogen phosphate), and the rate ratio of eluents A and B was 20:80. The flow rate was maintained at $1\,mL \cdot min^{-1}$, and the injection volume was 1.0 μL. The detection wavelength was 287 nm.

Three samples were conducted at each sampling site, and the concentrations of the samples were the average of the three measured values. All samples were analyzed in two groups of repeated trials. An external standard method was used to quantify the concentrations of the two antibiotics in water samples, and the correlation coefficients of standard curves were all greater than 99.8%. The recovery rates were identified prior to water sample pretreatment, and the results showed that they ranged from 80% to 105%, with relative standard deviations (RSDs) ranging from 5.2% to 10.3%. Through SPE enrichment, the detection limit of water samples was $0.001\,ng \cdot L^{-1}$.

## 2.4. Health Risk Assessment

In this study, health risks (*HQs*) were used to assess human health hazards. Exposure pathways of health risk assessment of antibiotics in the Yinma River Basin were divided into drinking water,

fish consumption, and skin exposure. The exposure doses from drinking water, fish consumption, and skin exposure were calculated using Equations (1), (2), and (3), respectively [21–28]:

$$ADD_D = \frac{MEC \times IR \times EF \times ED}{BW \times AT} \tag{1}$$

$$ADD_F = \frac{MEC \times BCF \times IR \times EF \times ED}{BW \times AT} \tag{2}$$

$$ADD_W = \frac{MEC \times S_A \times P_C \times ET \times ED}{BW \times AT} \tag{3}$$

where $ADD_D$ ($\mu g \cdot (Kg \cdot day^{-1})^{-1}$) is the exposure dosage through direct ingestion; $ADD_F$ ($\mu g \cdot (Kg \cdot day^{-1})^{-1}$) is the exposure dosage through fish consumption; $ADD_W$ ($\mu g \cdot (Kg \cdot day^{-1})^{-1}$) is the exposure dosage through skin exposure; $MEC$ ($\mu g \cdot L^{-1}$) is the measured environmental concentration; $IR$ ($L \cdot day^{-1}$ or $g \cdot d^{-1}$) is the rate of direct ingestion or fish consumption; $EF$ ($day \cdot year^{-1}$) is the exposure frequency; $ED$ (year) is the exposure duration over a lifetime; $BW$ (Kg) is the average body weight; $AT$ (day) is the average lifetime for non-carcinogens and carcinogens; $BCF$ ($L \cdot Kg^{-1}$) is the fish bioconcentration factor; $SA$ ($m^2$) is the skin surface area; $PC$ ($m \cdot h^{-1}$) is the coefficient of pollutant skin permeability; and $ET$ (h) is the skin exposure time in contaminants.

To evaluate comprehensive human health hazards, HQ was calculated as follows:

$$HQ = \frac{ADD}{RfD} = \frac{ADD_W + ADD_f + ADD_D}{RfD} \tag{4}$$

where $RfD$ ($\mu g \cdot (Kg \cdot day^{-1})^{-1}$) is the reference dosage of certain pollutants exposed through a certain exposure pathway. The $RfD$ is a reference value. If the exposure dose is lower than $RfD$, it may not cause harmful health effects, but if the exposure dose is higher than RfD, harmful health effects may be incurred (*Expose Factors Handbook of China Population*). These parameters, e.g., *IR*, *EF*, *ED*, *BW*, *AT*, *BCF*, *SA*, *PC*, *ET*, and *RfD*, were based on the *Expose Factors Handbook of China Population* and other literature [29]. In reference to the local situation of the Yinma River Basin, a questionnaire was used to survey the residents that were living in the sampling points (shown in the parameter acquisition and determination). If *HQs* values were less than 1, the chemicals were considered to be of little to no hazard; while *HQs* values greater than 1 indicated highly hazardous chemicals. When calculating the total risk of the two antibiotics, the corresponding risk values were first calculated separately and were then added.

In the calculation of health risks, according to the proportion of men and women in the sampling points, the weight method was used to calculate the health risks of residents in the sampling points. The formula used was as follows:

$$HQ = HQ_{male} \times P_{male} + HQ_{female} \times P_{female} \tag{5}$$

The health risk assessment parameters used in this study were obtained by the *Expose Factors Handbook of China Population* and the field survey, as shown in the following Tables 1 and 2.

**Table 1.** Daily average intake parameters of three types of exposure of residents in the Yinma River Basin, Northeast China.

| Parameter Name | | Value (mL·d$^{-1}$) | Parameter Name | | Value (mL·d$^{-1}$) | Parameter Name | | Value (g·d$^{-1}$) |
|---|---|---|---|---|---|---|---|---|
| IR (Drinking water) | August | | ET (skin exposure) | August | | IR (fish consumption) | August | |
| | | rural male 1854 | | | rural male 8 | | | urban area 40 |
| | | rural female 1628 | | | rural female 9 | | | |
| | | urban male 1913 | | | urban male 9 | | | reservoir area 35 |
| | | urban female 1774 | | | urban female 10 | | | |
| | | suburb male 1882 | | | suburb male 9 | | | other area 30 |
| | | suburb female 1700 | | | suburb female 9 | | | |
| | November | | | November | | | November | |
| | | rural male 1086 | | | rural male 3 | | | urban area 28 |
| | | rural female 994 | | | rural female 4 | | | |
| | | urban male 1375 | | | urban male 5 | | | reservoir area 23 |
| | | urban female 1264 | | | urban female 6 | | | |
| | | suburb male 1225 | | | suburb male 4 | | | other area 18 |
| | | suburb female 1127 | | | suburb female 5 | | | |

**Table 2.** Parameters used for health risk assessment in the Yinma River Basin, Northeast China.

| Parameter Name | Unit | Classification | Value |
|---|---|---|---|
| Exposure Frequency (*EF*) | d·year$^{-1}$ | | 365 |
| Exposure Duration Over a Lifetime (*ED*) | Year | male | 74.12 |
| | | female | 78.44 |
| Bioconcentration Factor (*BCF*) | L·Kg$^{-1}$ | | 3.2 |
| Skin Surface Area (*SA*) | m$^2$ | rural male | 1.7 |
| | | rural female | 1.6 |
| | | urban male | 1.8 |
| | | urban female | 1.6 |
| | | suburb male | 1.7 |
| | | suburb female | 1.6 |
| Coefficient of Pollutant Skin Permeability (*PC*) | m·h$^{-1}$ | | 0.02 |
| Average Body Weight (*BW*) | Kg | rural male | 66.1 |
| | | rural female | 59.8 |
| | | urban male | 70.6 |
| | | urban female | 62.4 |
| | | suburb male | 68.3 |
| | | suburb female | 61.1 |
| Average Lifetime (*AT*) | D | male | 27,053.8 |
| | | female | 28,630.6 |

The reference dosages (RfDs) of different pollutants in different exposure pathways were also assessed. The RfDs of the two target antibiotics under different exposure pathways used in this study are shown in Table 3.

**Table 3.** Exposure parameters of two target antibiotics under different exposure pathways [29,30].

| Antibiotics | Exposure Pathway | Value (µg·Kg$^{-1}$·d$^{-1}$) |
|---|---|---|
| NOR | skin exposure | 380 |
| | direct consumption | 190 |
| CIP | skin exposure | 3.2 |
| | direct consumption | 1.6 |

## 3. Results and Discussion

### 3.1. Spatiotemporal Distribution of Antibiotics

Mean (± standard deviation) concentrations of norfloxacin and ciprofloxacin of the Yitong tributary in the wet and dry season (22 August 2015 and 1 November 2015) were 64.669 ± 12.046 and 3.329 ± 1.412 ng·L$^{-1}$ in the wet season and 57.42 ± 18.465 and 3.138 ± 1.229 ng·L$^{-1}$ in the dry season. For the Yinma tributary, mean concentrations of NOR and CIP in the wet and dry season were 59.068 ± 14.170 and 3.772 ± 0.750 ng·L$^{-1}$ in the wet season and 54.043 ± 13.230 and 3.071 ± 0.578 ng·L$^{-1}$ in the dry season (Table 4). The mean concentration of NOR in the wet season was higher than in the dry season, and CIP was the same.

Based on the statistical test on the antibiotic levels between the wet and dry season, we found no seasonal difference in NOR but strong seasonal differences in CIP in the Yinma River. The concentration of CIP was higher in the wet season than in the dry season across the river basin. We also found that the Yitong tributary showed a more prevalent seasonal difference in CIP than the Yinma tributary.

**Table 4.** Norfloxacin (NOR) and ciprofloxacin (CIP) concentrations (ng·L$^{-1}$) in the Yitong tributary (YT), Yinma tributary (YM), and other smaller tributaries (Other) in the Yinma River Basin, Northeast China, during the wet and dry seasons of 2015.

| Site | Wet | | Dry | | Mean | |
|---|---|---|---|---|---|---|
| | NOR-Wet | CIP-Wet | NOR-Dry | CIP-Dry | NOR-Mean | CIP-Mean |
| YT 1 | 70.611 | 2.04 | 51.719 | 1.989 | 61.164 | 2.015 |
| YT 2 | 41.518 | 1.221 | 35.335 | 1.343 | 38.427 | 1.282 |
| YT 3 | 64.497 | 4.491 | 75.034 | 4.276 | 69.766 | 4.384 |
| YT 4 | 72.099 | 4.514 | 82.787 | 4.252 | 77.443 | 4.383 |
| YT 5 | 74.618 | 4.381 | 42.226 | 3.83 | 58.422 | 4.105 |
| Mean | 64.669 | 3.329 | 57.42 | 3.138 | 61.044 | 3.234 |
| SD | 12.046 | 1.412 | 18.465 | 1.229 | 13.141 | 1.319 |
| YM 1 | 48.211 | 3.415 | 41.494 | 3.245 | 44.852 | 3.33 |
| YM 2 | 56.913 | 3.245 | 39.437 | 2.569 | 48.175 | 2.907 |
| YM 3 | 60.281 | 3.18 | 57.227 | 2.53 | 58.754 | 2.855 |
| YM 4 | 36.781 | 3.302 | 77.166 | 2.861 | 56.973 | 3.082 |
| YM 5 | 74.838 | 4.236 | 46.273 | 2.977 | 60.556 | 3.607 |
| YM 6 | 77.382 | 5.251 | 62.658 | 4.243 | 70.02 | 4.747 |
| Mean | 59.068 | 3.772 | 54.043 | 3.071 | 56.555 | 3.421 |
| SD | 14.170 | 0.750 | 13.23 | 0.578 | 8.261 | 0.646 |
| S 1 | 43.85 | 2.582 | 51.024 | 2.831 | 47.437 | 2.707 |
| S 2 | 74.26 | 3.279 | 78.765 | 3.388 | 76.512 | 3.333 |
| S 3 | 56.59 | 2.495 | 53.22 | 2.335 | 54.905 | 2.415 |
| S 4 | 41.857 | 3.174 | 57.028 | 2.484 | 49.443 | 2.829 |
| S 5 | 63.183 | 3.913 | 49.185 | 3.426 | 56.184 | 3.669 |
| S 6 | 80.575 | 3.975 | 75.823 | 3.964 | 78.199 | 3.969 |

Longitudinally, we found a clear difference in the concentration of NOR and CIP between the upstream and downstream Yinma River (Figure 3). There appeared to be an increasing trend of the antibiotic concentration from upstream to downstream. The mean concentrations of NOR in the Yitong and Yinma tributaries were 61.044 ± 13.141 and 56.555 ± 8.261 ng·L$^{-1}$, fluctuating from 77.443 to 38.426 ng·L$^{-1}$ and 70.020 to 44.852 ng·L$^{-1}$, respectively. The mean concentrations of CIP in the Yitong and Yinma tributaries were 3.234 ± 1.319 and 3.421 ± 0.646 ng·L$^{-1}$, fluctuating from 4.383 to 1.282 ng·L$^{-1}$ and 4.747 to 2.855 ng·L$^{-1}$, respectively.

Intensive livestock production and residential areas were major pathway for antibiotics usage. A previous study reported that a large amount of antibiotics could remain in the soil [31]. During the wet season, large quantities of antibiotics could have been released into rivers by surface runoff and leaching. Therefore, antibiotics had a higher concentration in the wet season. In intensive agriculture, antibiotics are used as feed supplements [32] throughout the year. This is the case with the Yinma tributary area, which is predominantly in livestock production; Hence, the antibiotic level in the tributary did not show a significant difference between the wet season and dry season.

Although antibiotics are easily degradable, they can still persist in the environment in a strong relationship with typical sources of pollution [9]. Two counties within the Yinma River Basin, Nongan and Dehui, are areas with intensive livestock production. Agriculture and industry are dominant in the upper Yinma River Basin, making a large contribution to antibiotic pollution.

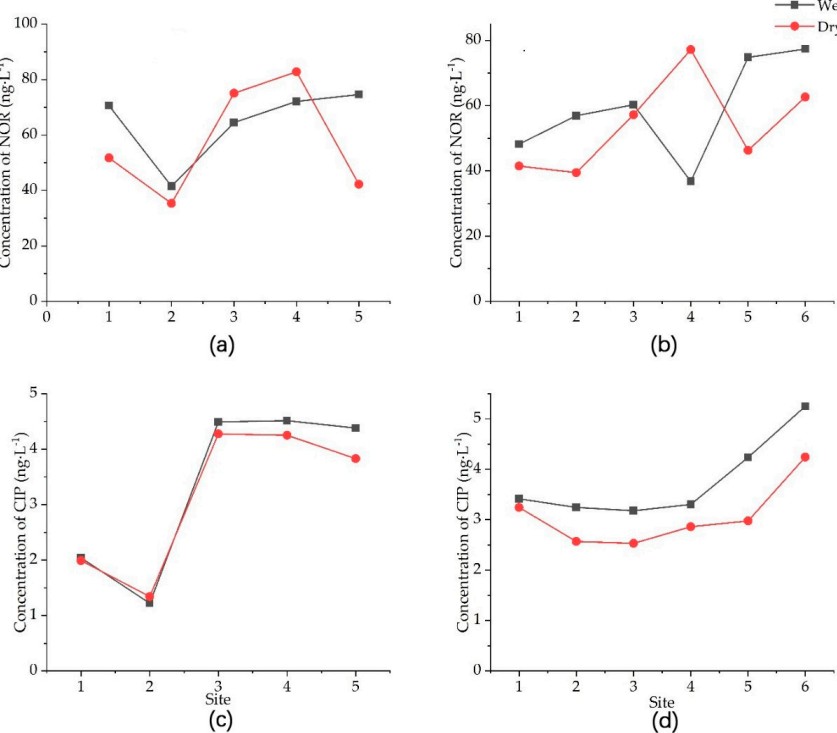

**Figure 3.** Concentration of two popularly used antibiotics, norfloxacin (NOR) and ciprofloxacin (CIP), in the Yitong (**a**,**c**) and Yinma (**b**,**d**) tributaries of the Yinma River in Northeast China during the 2015 wet and dry seasons.

### 3.2. Relationship between Environmental and Social Factor and Antibiotics

Environmental and social factors affecting the level of antibiotics in the Yinma River included water quality, land use, population, GDP, and slope. Since hydrological conditions are significantly affected by seasons, only two typical water seasons (i.e., dry and wet) were chosen for this study.

Regarding water quality, the concentration of NOR showed a correlation with DO and DOC in the wet season, and had a strong correlation with DO. In the dry season, no statistically significant relationship between NOR and water quality was found. CIP had a correlation with ammonia nitrogen and DOC in both the wet and dry season. There was no statistically significant difference between the other water quality index and NOR or CIP (Table 5). A study by Polubesovaa and others showed that some functional groups in DOC can enhance antibiotics migration by cotransport [33]. In this study, antibiotics had a higher concentration in the wet season, resulting in a positive correlation with DOC.

There was a positive correlation between the NOR and CIP concentrations in the Yinma River water and the local topographical slope. Through the analysis, it was found that regarding the antibiotics concentration at each land-use type, antibiotics in wetland and forest areas displayed maximum concentrations, while low-level concentrations were found in paddy field areas. When the river flows through wetland with a low velocity, pollution will stay and collect in this area. Hence, wetland areas may be a contamination source of antibiotics with high concentrations. In areas with a higher slope, the land cover is usually forest, as previously reported. Additionally, antibiotics had high concentrations in forest areas. Therefore, the combination of slope and land-use type affected the distribution of antibiotics in the river basin.

Two previous studies reported high concentrations of antibiotics in urban and densely populated areas [7,34]. In this study, however, we found a negative relation between the antibiotics and the population and GDP in the Yinma River (Figure 4), indicating a stronger influence of livestock production in rural areas. The rural area of the Yinma River Basin is dominated by livestock production

and intensive agriculture and, therefore, the distribution of antibiotics in the Yinma River is reflective of the land use.

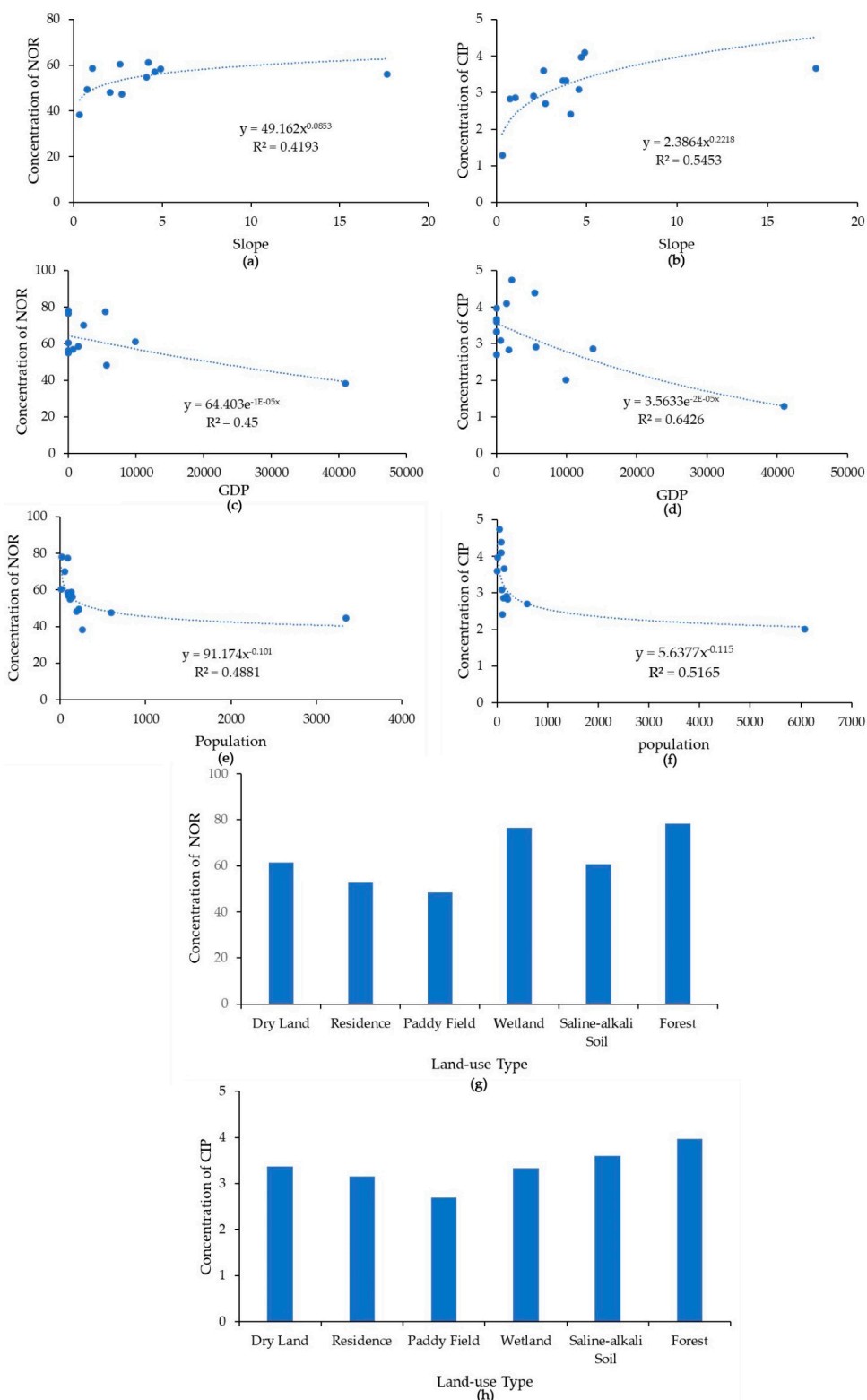

**Figure 4.** (**a**) Relationships of norfloxacin (NOR) and ciprofloxacin (CIP) concentrations (ng·L$^{-1}$) in the Yinma River, Northeast China, with slope in degrees (**a**,**b**), with GDP in RMB (**c**,**d**), population (person) (**e**,**f**),and land use type (**g**,**h**).

**Table 5.** Pearson coefficient correlation of two popularly used antibiotics, norfloxacin (NOR) and ciprofloxacin (CIP), and other water quality parameters in the 2015 wet and dry seasons in the Yinma River, Northeast China.

| Water Quality | Wet | | Dry | |
|---|---|---|---|---|
| | NOR | CIP | NOR | CIP |
| Temperature (°C) | 0.0461 | 0.2485 | −0.0808 | −0.1179 |
| PH | −0.005 | 0.1996 | 0.0711 | −0.0994 |
| Dissolved Oxygen (mg·L$^{-1}$) | 0.6438** | −0.4156 | −0.2315 | −0.2993 |
| Free chlorine (mg/L) | 0.0141 | 0.2583 | 0.2888 | 0.2526 |
| Ammonia Nitrogen (mg/L) | 0.269 | 0.5159* | 0.0439 | 0.5344* |
| COD (mg·L$^{-1}$) | 0.4894* | 0.5442* | 0.4261 | 0.6333** |

** Correlation is significant at the 0.01 level. * Correlation is significant at the 0.05 level.

*3.3. Health Risk Assessment of Antibiotics*

The mean health risk of norfloxacin and ciprofloxacin in the Yinma River for the wet and dry seasons was 0.638 ± 0.189 and 0.354 ± 0.110, respectively, fluctuating from 0.955 to 0.222 and from 0.584 to 0.146. NOR had a higher health risk than CIP for residents living in the Yinma River Basin. The *t*-test revealed a significant seasonal difference ($p = 4.79 \times 10^{-6}$, $p < 0.0001$) in health risks. During the wet season, both NOR and CIP showed a higher risk than during the dry season (Figure 5). During both the wet and dry seasons, the health risk in the low river basin was generally higher than in the upper river basin (Figure 5).

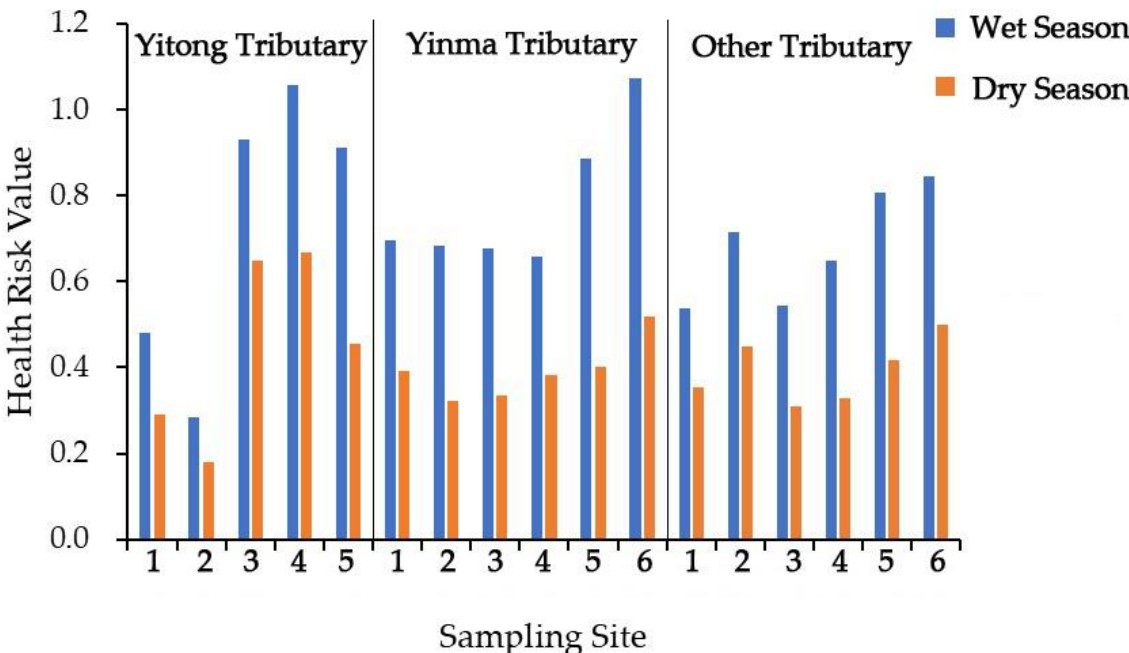

**Figure 5.** Health risk value of two antibiotics (norfloxacin and ciprofloxacin) found in stream waters during the 2015 wet and dry seasons in the Yinma River, Northeast China.

Through analysis of the antibiotics' health risk in different areas in the Yinma River Basin, contrary to the distribution of the antibiotic concentration, the health risk in urban areas was higher than suburban and rural areas (Figure 6). At sampling sites YT 3, the health risk value was significantly higher than other points. This is because of human life habits; the dose of water for human ingestion is different depending on the area, so the risk is different.

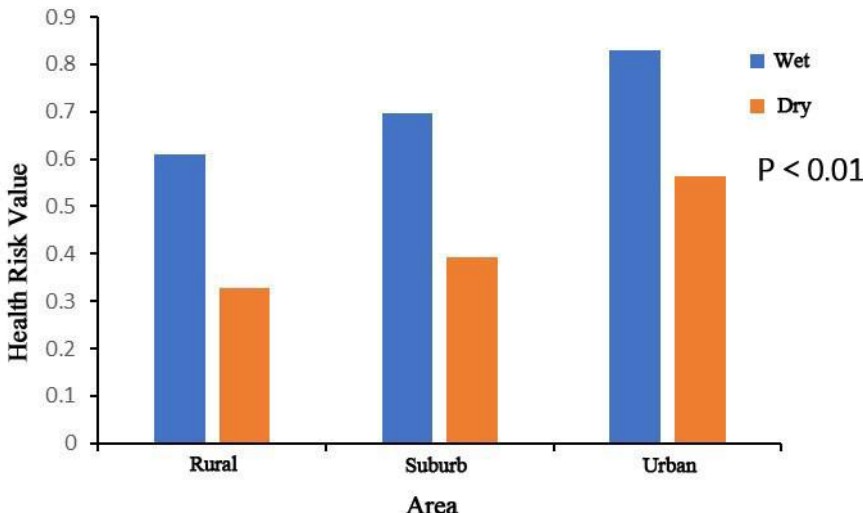

**Figure 6.** Health risk value of two antibiotics (norfloxacin and ciprofloxacin) found in streams draining different land use areas in the Yinma River Basin, Northeast China.

Therefore, in health risk management, it is not rigorous to only focus on the concentration of pollutions. According to the risk distribution in different areas, we can make corresponding buffers to avoid risks [35]. For the Yinma River Basin, health risk buffers of antibiotics are show in Figure 7.

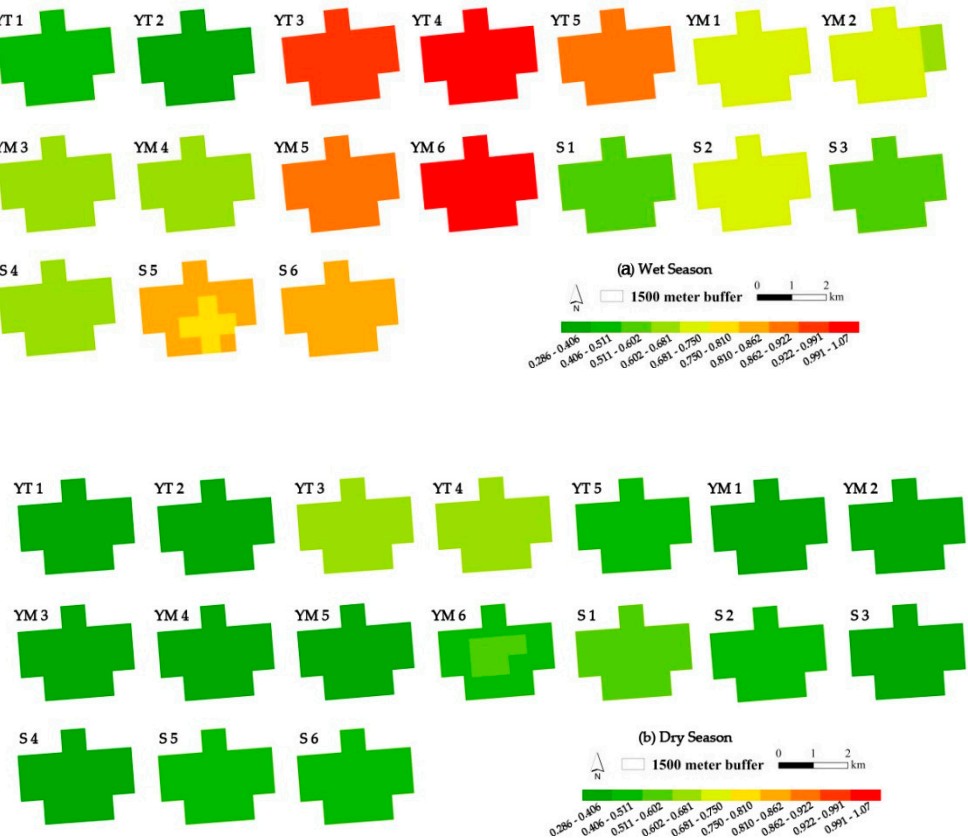

**Figure 7.** Health risk buffers of antibiotics in the Yinma River Basin. The buffers of 1500 m in the wet (**a**) and dry (**b**) season in the Yinma River Basin were established by ArcGIS 10.0.

The establishment of buffers was completed to reduce the health risk of the residents affected by pollutants, but also is more targeted to pollution control of the river basin. Therefore, the establishment

of buffers was shown to be an effective risk method for managing and predicting the health risk of residents in the basin.

## 4. Conclusions

This study found the presence of two antibiotics, ciprofloxacin and norfloxacin, in waterways of a headwater area in Northeast China, indicating the effect of intensive livestock production on water quality and health risk. Overall, norfloxacin showed a much higher level ($-59$ ng L$^{-1}$) compared to ciprofloxacin ($-3.2$ ng L$^{-1}$) in the waterways, indicating a greater use of the first in livestock production. Rainfall can cause a higher level of pollution of both antibiotics across the river basin, while topography and land use contribute to the spatial distribution of the antibiotics. The level of antibiotics was higher in rural areas, especially forested and wetland areas where livestock typically graze. Both antibiotics also showed a higher level in downstream waters. However, the health risk of antibiotics is determined by the physical condition and lifestyle of the residents in the river basin, hence the urban area presents a higher vulnerability than the rural area. Therefore, establishing stream buffer zones may be an effective approach for antibiotic risk management in river basins dominated by intensive livestock production.

**Supplementary Materials:** The following are available online at http://www.mdpi.com/2073-4441/11/10/2006/s1, Figure S1: The land use distribution and layout of sampling sites in the Yinma River Basin, Figure S2: (a). The HQ distribution in wet season in the Yinma River Basin, (b). The HQ distribution in dry season in the Yinma River Basin, Table S1: The three parallel experiments concentration of antibiotics in two seasons of the sampling sites in the Yinma River Basin, Table S2: Various environmental factors of the sampling sites in the Yinma River Basin, Table S3: The water quality in two seasons of the sampling sites in the Yinma River Basin, Table S4: The health risk in two antibiotics in two seasons in the Yinma River Basin.

**Author Contributions:** H.J. and S.L. conceived and designed the study; H.J. performed the field sampling, measurements and data analysis; Y.J.X. and H.J. wrote and revised the manuscript; Y.J.X. provided conceptual overview of the manuscript preparation, writing and revision. G.Z. and J.Z. provided funding support for this study.

**Funding:** This study was supported by the National Key R&D Program of China. Grant: 2017YFC0406003 and the National Science and Technology Development Plan, project of Jinlin Province, China. Grant No. 20190303081SF.

**Acknowledgments:** We thank the Northeast Normal University for providing laboratory analyses. Sincere thanks also go to two anonymous reviewers for their very helpful comments and suggestions, which have helped us clarify and improve the manuscript.

**Conflicts of Interest:** The authors declare no conflict of interest.

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
