# Peer review of "Intensive Livestock Production Causing Antibiotic Pollution in the Yinma River of Northeast China"

_water, doi:10.3390/w11102006_

Round 1
Reviewer 1 Report
The work raises the topic of antibiotic levels in the aquatic environment. A very extensive research area is planned in the river basin of two rivers. An attempt was also made to determine the significance of the impact of compounds on health.
Main comments
Please provide research hypotheses so that the work is not just a report. One of the important elements of the assessed work is a comparison of the dry and wet period, but the characteristics of these periods are missing. In addition, the hydrological characteristics of both rivers have been omitted and these conditions determine the amount of antibiotics in the water. Demonstrating the impact of urban areas on the basis of one point is methodically preliminary. Despite the large sampling, the method of selecting places is unconvincing and hinders the use of appropriate methods of analysis, e.g. PERMANOVA. It has not been explained why two antibiotics, norfloxacin (NOR) and ciprofloxacin (CIP) were selected, and the indication that they are popular is insufficient.
There is no repetition. The description shows that 2 samples were taken at each site. This determines the measurement errors that are difficult to determine and affect the entire logical argument of the work. This is especially evident in figure 3.
Comparing both antibiotics without knowing their characteristics at this rate of decomposition can give wrong information.
The sources of information on the amount of antibiotics supplied by fish consumption are incomprehensible. Is this the author's data or taken from the literature. Table 3 is very mysterious.
Simple data sets in bar charts are not convincing (figs. 5 and 6) . Why was the significance of these data differences not being determined in pair tests? That the value is higher or lower need not be importent to scientific considerations.
What buffer zones do the authors mention? This is incomprehensible and the indicated drawing does not help explain the intentions of the authors. This is the most important job achievement and should not be objectionable and should be written with a clear and precise.
Detal comments:
line 49 maybe better information is value of concetration
line 68 hydrological and climatic conditions seem to be the main element
What are the units in Figures 2a, c and d? The scale should be in whole values. Delete repetitive elements and descriptions that will appear in the title. The quality of this graphic is low and requires better quality. The description contains too many elements of the methodology. Please correct the title of the drawing.
line 134-135 is this real information? The area is significant and maintaining such an interval is puzzling.
What specific analytical equipment was used, brand, country ect.?
lines 269-278 The correlation coefficient is definitely not enough in terms of statistical analyzes.
Author Response
The work raises the topic of antibiotic levels in the aquatic environment. A very extensive research area is planned in the river basin of two rivers. An attempt was also made to determine the significance of the impact of compounds on health.
[Authors’ response] Thank you so much for taking time to review our manuscript. We greatly appreciate you for your comments and suggestions, all of which have been taking into account into the revised manuscript, as described below.
Main comments
Please provide research hypotheses so that the work is not just a report.
[Authors’ response]: Thank you for the suggestion.
We have added a hypothesis in the revised manuscript according (see lines 70-73).
One of the important elements of the assessed work is a comparison of the dry and wet period, but the characteristics of these periods are missing.
[Authors’ response] We have added explanation to clarify the difference between the two seasons in the revised manuscript (see lines 89-94).
In addition, the hydrological characteristics of both rivers have been omitted and these conditions determine the amount of antibiotics in the water.
[Authors’ response] Good point. We have added text to explain hydrological conditions of the Yinma and Yitong tributaries in the revised manuscript (lines 86-94).
Demonstrating the impact of urban areas on the basis of one point is methodically preliminary.
[Authors’ response] The sampling site selected as a representative of the urban areas in this study is the sewage outlet at the most downstream of the city. Therefore, this sampling site is used to represent the pollution level of urban areas, as shown in the appendix figure 1.
Despite the large sampling, the method of selecting places is unconvincing and hinders the use of appropriate methods of analysis.
[Authors’ response] We agree that the sampling has some limitation for statistical testing. Nonetheless, the results gained from the study provide plausible indication for livestock production effects on downstream antibiotic pollution. We have added explanation in the revised manuscript (lines 132-135), as well as specifically indicated in the appendix figure 1.
It has not been explained why two antibiotics, norfloxacin (NOR) and ciprofloxacin (CIP) were selected, and the indication that they are popular is insufficient.
[Authors’ response] Your point is well taken. We have added text to explain why the two antibiotics were chosen in this study, see lines 103-105 in the revised manuscript.
There is no repetition. The description shows that 2 samples were taken at each site. This determines the measurement errors that are difficult to determine and affect the entire logical argument of the work. This is especially evident in figure 3.
[Authors’ response] Thank you for your careful reading. We apologize for the unclarity (a mistake) in the previous manuscript. In fact, we collected three samples in the field and we have added the following to explain (line 175-176): “Three samples were taken at each sampling site, and the concentrations of the samples were measured and averaged for the sampling site.”. These data of antibiotics are shown in the appendix Table 1.
Comparing both antibiotics without knowing their characteristics at this rate of decomposition can give wrong information.
[Authors’ response] Thank you for your comment. We have revised the manuscript, not to conduct a comparative study of the two antibiotics, but to analyze the distribution and occurrence rules of antibiotics from the perspective of season and space. The reason why CIP and NOR are selected in this study is that they are widely used and have caused concerns about contamination in the research area.
The sources of information on the amount of antibiotics supplied by fish consumption are incomprehensible. Is this the author's data or taken from the literature. Table 3 is very mysterious.
[Authors’ response] We have added data sources of Table 1 in the manuscript according to your comment, lines 213, and the sources of Table 3 of this paper are literatures, line 235.
Simple data sets in bar charts are not convincing (figs. 5 and 6) . Why was the significance of these data differences not being determined in pair tests? That the value is higher or lower need not be importent to scientific considerations.
[Authors’ response] Thank you for your suggestion. We have now added statistical analysis of the risk assessment according (see line 322 in the revised manuscript).
What buffer zones do the authors mention? This is incomprehensible and the indicated drawing does not help explain the intentions of the authors. This is the most important job achievement and should not be objectionable and should be written with a clear and precise.
[Authors’ response] We appreciate your honesty in your critique of our results section. We have described the acquisition of buffer zone in the manuscript in detail, as shown in the appendix.
Detal comments:
line 49 maybe better information is value of concetration.
[Authors’ response] As suggested, we added ppt for the concentration.
line 68 hydrological and climatic conditions seem to be the main element.
[Authors’ response] Thanks and we added them in the revised manuscript.
What are the units in Figures 2a, c and d? The scale should be in whole values. Delete repetitive elements and descriptions that will appear in the title. The quality of this graphic is low and requires better quality. The description contains too many elements of the methodology. Please correct the title of the drawing.
[Authors’ response] We have redone the figures and hope the changes meet your expectations (line 112-125).
line 134-135 is this real information? The area is significant and maintaining such an interval is puzzling.
[Authors’ response] As there may be accidental errors in the sampling procedure, three water samples from upstream and downstream 200m are collected for mixing to eliminate accidental errors. And I have described it in detail in the manuscript according to your comment, see lines 142.
What specific analytical equipment was used, brand, country ect.?
[Authors’ response] Thank you for catching this. We now added information about the equipment (line 164).
lines 269-278 The correlation coefficient is definitely not enough in terms of statistical analyzes.
[Authors’ response] The concentration distribution of antibiotics is jointly affected by multiple factors in the whole basin, so the correlation coefficient of single environmental factor on its influence is not particularly large, but still shows a certain degree of influence and correlation.
Again, thank you very much for your helpful comments and suggestions.
Reviewer 2 Report
The manuscript presents the results of investigating antibiotic occurrence in waterways of Northeast China. The Authors analyse spatiotemporal distribution and correlation of antibiotics with some environmental factors but also estimate health risk. Achieved results are interested for a broader community and merit to be published in the Water journal. The general structure of the manuscript is correct but it requires some major revision.
First of all the terminology used in hydrology should be applied appropriately. Please consider to remove from the title ‘a headwater area in’. If the Yinma River Basin is studied, it means that Yinma is the main river which drain the whole basin. All other rivers are tributaries. Therefore the Authors should not call Yinma as ‘tributary’. Please verify this throughout the manuscript. I would also suggest to divide the basin into sub-basins (catchments) to perform spatial analysis of sampling sites. It is unclear how environmental or social factors e.g. slope or population were assigned to sampling points? Moreover the results of health risk should be clarified. Please consider to add in Appendix the table with measurements of parameters listed in Table 7 (L279) and results of health risk calculations.
The specific comments to improve the manuscript are as follow:
L28 What the Authors mean by ‘chlorine consumption’? L49 Please explain PPT. L79 Please rewrite the sentence as it seems that Yinma is a tributary of itself. L81 Please consider to use word ‘confluence’ instead of ‘merge and flow together’ Is it true? They do not mix for 20 km? 89 Please add on the right hand side map the names of the rivers and sub-catchments boundaries. L95 Replace ‘Capitol’ with ‘capital’. L99 Please add the noun e.g ‘land’ at the end. L103 to 116 should not be part of figure’s title. Please make it part of the paragraph. Ward ‘date’ is misused here. Please replace it with ‘data’. Please remove ‘each 1 km×1 km raster data represents the GDP within this square kilometer’ as it was already written ‘1 km×1 km raster data’. L125 Please write explicitly how many samples in total were analysed? L170 Please clarify if two antibiotics are treated separately or summed up to calculate health risk. Please provide the reference where all the equations come from. L2008 Provide reference of ‘statistical yearbook’. L212 Please add full names of parameters in first column. L214-216 Please rewrite. L220 This paragraph should be expanded or removed. L232-233 Please rewrite the sentence as it is true only for CIP in YM. L234 Please renumber all tables hereafter. Consider to add column in Table 6 with number of samples. L244 Provide correct fumbers of refered figures. L246 and L248 Wrong numbers? L260-262 Not true. The difference is higher in YM than YT. L279 It should be ‘Pearson correlation coefficient‘. Explain the meaning of * and ** as well as ‘Free chlorine consumption’. Provide unit for temperature and Ammonia Nitrogen. L298 Not clear what area was considered for each sampling site? I would recommend to use sub-catchments. Provide units for Slope, GDP and Population. L300-301 Instead ‘c and d’ should be ‘g and h’ and the other way around. L317 Unclear which sites belong to which category. L325 What is the meaning of shapes on Figure 7? Some background information about the buffer and its definition is missing. Explain the role of ArcGIS and which tools where used.Author Response
The manuscript presents the results of investigating antibiotic occurrence in waterways of Northeast China. The Authors analyse spatiotemporal distribution and correlation of antibiotics with some environmental factors but also estimate health risk. Achieved results are interested for a broader community and merit to be published in the Water journal. The general structure of the manuscript is correct but it requires some major revision.
[Authors’ response] Thank you for your support. We greatly appreciate your time in thoroughly evaluating our manuscript and providing helpful comments and suggestions, all of which have been taken into account while we were revising the manuscript.
First of all the terminology used in hydrology should be applied appropriately. Please consider to remove from the title ‘a headwater area in’. If the Yinma River Basin is studied, it means that Yinma is the main river which drain the whole basin. All other rivers are tributaries. Therefore the Authors should not call Yinma as ‘tributary’. Please verify this throughout the manuscript.
[Authors’ response] Thanks for the suggestions. We have modified the title to “ Intensive livestock production causing antibiotic pollution in the Yinma River of Northeast China.” We have added explanations in the revised manuscript to clarify the definition of the Yinma and Yitong tributaries (lines 79-82).
I would also suggest to divide the basin into sub-basins (catchments) to perform spatial analysis of sampling sites. It is unclear how environmental or social factors e.g. slope or population were assigned to sampling points?
[Authors’ responses] Your point is well taken. Because the Yinma River Basin is comparably not vast and the number of our sampling sites is limited for meaningful spatial and/or statistical analysis, we did not treat the two tributary basins separately. To incorporate your suggestion, we have included the data of all environmental factors in the appendix in the revised manuscript (see appendix Table 2 and Table 3).
The results of health risk should be clarified. Please consider to add in Appendix the table with measurements of parameters listed in Table 7 (L279) and results of health risk calculations.
[Authors’ responses] As suggested, we have included the data of water quality parameters and health risk values in the appendix according to your comment, as shown in the appendix table 3 and table 4.
The specific comments to improve the manuscript are as follow:
L28 What the Authors mean by ‘chlorine consumption’?
[Authors’ responses] Free chlorine Consumption means characterization of Free chloride concentration in the manuscript. We have clarified it (lines 28 and 297).
L49 Please explain PPT.
[Authors’ responses] We have described the PPT (ng/L) in the revised manuscript (line 48).
L79 Please rewrite the sentence as it seems that Yinma is a tributary of itself.
[Authors’ responses] We have clarified the Yinma tributary and the Yinma River Basin in the revised manuscript (lines 70-73).
L81 Please consider to use word ‘confluence’ instead of ‘merge and flow together’ Is it true? They do not mix for 20 km?
[Authors’ responses] As suggested, we have rephrased the sentence. (lines 84-86).
89 Please add on the right hand side map the names of the rivers and sub-catchments boundaries.
[Authors’ response] As suggested, the Yinma tributary and Yitong tributary were marked on the map of the study area (line 96).
L95 Replace ‘Capitol’ with ‘capital’.
[Authors’ responses] We replaced ‘Capitol’ with ‘capital’ (line 100).
L99 Please add the noun e.g ‘land’ at the end.
[Authors’ response] We added the noun (line 106).
L103 to 116 should not be part of figure’s title. Please make it part of the paragraph. Ward ‘date’ is misused here. Please replace it with ‘data’. Please remove ‘each 1 km×1 km raster data represents the GDP within this square kilometer’ as it was already written ‘1 km×1 km raster data’.
[Authors’ response] As suggested, we put the text into a paragraph in the revised manuscript (line 112-125).
L125 Please write explicitly how many samples in total were analysed?
[Authors’ response] A total of 102 samples were analyzed and we explained it in the revised manuscript (lines 137-138).
L170 Please clarify if two antibiotics are treated separately or summed up to calculate health risk. Please provide the reference where all the equations come from.
[Authors’ response] In this study, the health risks of antibiotics were calculated separately and then added up. We have marked the source of the formula according to your comment, line 190.
L208 Provide reference of ‘statistical yearbook’.
[Authors’ response] We added “the Expose Factors Handbook of China Population” as the statistical yearbook (line 224-225).
L212 Please add full names of parameters in first column.
[Authors’ response] We have added full names of parameters in first column according to your comment, line 229-230.
L214-216 Please rewrite.
[Authors’ response] We have rewritten the sentence (line 232).
L220 This paragraph should be expanded or removed.
[Authors’ response] We have removed '2.5 Statistical analysis'.
L232-233 Please rewritten the sentence as it is true only for CIP in YM.
[Authors’ response] We have rewritten the sentence (line 245).
L234 Please renumber all tables hereafter.
[Authors’ response] Done.
Consider to add column in Table 6 with number of samples.
[Authors’ response] As suggested we added a column in Table 6 with the number of samples (line 248-249).
L244 Provide correct numbers of figures.
[Authors’ response] Done (line 256).
L246 and L248 Wrong numbers?
[Authors’ response] We have rewritten this part (line 258-261).
L260-262 Not true. The difference is higher in YM than YT.
[Authors’ response] Thank you for your careful reading. We have corrected it (line 273).
L279 It should be ‘Pearson correlation coefficient’. Explain the meaning of * and ** as well as ‘Free chlorine consumption’. Provide unit for temperature and Ammonia Nitrogen.
[Authors’ response] The mean of ‘*’ is correlation is significant at the 0.05 level, the mean of ‘**’ is correlation is significant at the 0.01 level, and the Free chlorine Consumption means characterization of free chloride ion concentration, I have described it in the manuscript according to your comment, line 295-297.
L298 Not clear what area was considered for each sampling site? I would recommend to use sub-catchments.
[Authors’ response] Thank you very much again for your comment. Since the Yinma River Basin is not a vast research area and the number of sampling points is not large enough, the number of samples for statistical analysis is too small if they are divided into sub-catchments for research, resulting may in a large error. However, I have added the data of all environmental factors in the appendix of the manuscript according to your comment, as shown in the appendix table 2 and table 3.
Provide units for Slope, GDP and Population.
[Authors’ response] Done (line 315-316)
L300-301 Instead ‘c and d’ should be ‘g and h’ and the other way around.
[Authors’ response] The suggested change has been done (line 315-316).
L317 Unclear which sites belong to which category.
[Authors’ response] Sorry for the confusion. We have listed the classification of each sampling point (lines 132-135).
L325 What is the meaning of shapes on Figure 7? Some background information about the buffer and its definition is missing. Explain the role of ArcGIS and which tools where used.
[Authors’ response] We have described the acquisition of buffer zone in the manuscript in detail according to your comment, as shown in the appendix.
Thank you very much again for your helpful comments and suggestion, have helped us improve the quality of this manuscript.
Round 2
Reviewer 1 Report
Despite the corrections made, the work requires further refinement. Some corrections did not bring the expected increase in understanding of the text. There are also new fragments that cannot be understood.
Detailed comments:
Fig. 2 no units
Line 585 is the population really an environment variable? You cannot use abbreviations, e.g. GDP
Line 639 to correct the data concerning the apparatus is enough to provide the type and country
Lines 650-651 please correct the style of this sentence
Unify the record of the weight unit Kg or kg
Table 4 reverse the tables and supplement each value with SD. Redesign tables and remove duplicate information. Can put Wet and Dry side by side. The average values ​​should be separated
Fig. 3 removing the descriptions from the drawing. Transfer the river names for signature. The frame-mustache chart should be replaced
Are there significant differences between dry and vet
line 757 I don't understand this change
Table 5 change the description of the reaction abbreviation. Where are the missing units?
Fig. 4 missing units on the Y axis
line 806 improve the record of significance of differences, e.g. P <0.0001
Figs. 5 and 6, no units on the Y axis. This combination is hard to read. Please strengthen the message with significance tests and parse analysis
In-depth statistical analysis is required at work. Are the results from the wet and dry periods statistically significant. How it looks in relation to both catchments. There are no answers to these questions.
Reviewer 2 Report
As this is my second chance to review this manuscript I must say that I am not fully confident with the replies to some of my first comments.
[Authors’ responses] Free chlorine Consumption means characterization of Free chloride concentration in the manuscript. We have clarified it (lines 28 and 297).
I still have doubts what the authors have measured? In the Appendix (Table 3) is written ‘Free chlorine consumption volume (ml)’. Please provide clear definition of this parameter.
[Authors’ responses] We have described the PPT (ng/L) in the revised manuscript (line 48).
The sentence does not explain the PPT sufficiently.
[Authors’ response] As suggested, the Yinma tributary and Yitong tributary were marked on the map of the study area (line 96).
There must be misspelling on the map of word ‘Yitong’.
[Authors’ response] As suggested, we put the text into a paragraph in the revised manuscript (line 112-125).
The authors did place the text into the paragraph but didn’t edit it. There are still multiple occurrences of word ‘date’. What is the use to write that 1 km x 1 km is the square kilometer?
[Authors’ response] We have rewritten the sentence (line 232).
L232 still contains a sentence with no verb.
L245 There should be reference to Table 4 not 1.
L293 Please add ‘of’ after ‘coefficient’.
[Authors’ response] We have described the acquisition of buffer zone in the manuscript in detail according to your comment, as shown in the appendix.
The authors provided supplementary materials but they did not write any reference to this materials. There is still no explanation what does the shape of a single puzzle on Figure 7 mean?
